# Mutations in Epigenetic Regulation Genes in Gastric Cancer

**DOI:** 10.3390/cancers13184586

**Published:** 2021-09-13

**Authors:** Marina V. Nemtsova, Alexey I. Kalinkin, Ekaterina B. Kuznetsova, Irina V. Bure, Ekaterina A. Alekseeva, Igor I. Bykov, Tatiana V. Khorobrykh, Dmitry S. Mikhaylenko, Alexander S. Tanas, Vladimir V. Strelnikov

**Affiliations:** 1Laboratory of Medical Genetics, I.M. Sechenov First Moscow State Medical University, 119991 Moscow, Russia; nemtsova_m_v@mail.ru (M.V.N.); kuznetsova.k@bk.ru (E.B.K.); bureira@mail.ru (I.V.B.); ekater.alekseeva@gmail.com (E.A.A.); dimserg@mail.ru (D.S.M.); 2Laboratory of Epigenetics, Research Centre for Medical Genetics, 115522 Moscow, Russia; akalinkin@epigenetic.ru (A.I.K.); atanas@med-gen.ru (A.S.T.); 3Department No. 1, Medical Faculty, Faculty Surgery, I.M. Sechenov First Moscow State Medical University, 119991 Moscow, Russia; igor-vr@mail.ru (I.I.B.); horobryh68@list.ru (T.V.K.)

**Keywords:** gastric cancer, epigenetic regulation genes, somatic mutations, molecular genetic markers

## Abstract

**Simple Summary:**

Epigenetic mechanisms, such as DNA methylation/demethylation, covalent modifications of histone proteins, and chromatin remodeling, create specific patterns of gene expression. Epigenetic deregulations are associated with oncogenesis, relapse of the disease and metastases, and can serve as a useful clinical marker. We assessed the clinical relevance of integrity of the genes coding for epigenetic regulator proteins by mutational profiling of 25 genes in 135 gastric cancer (GC) samples. Overall, mutations in the epigenetic regulation genes were found to be significantly associated with reduced overall survival of patients in the group with metastases and in the group with tumors with signet ring cells. We have also discovered mutual exclusivity of somatic mutations in the *KMT2D*, *KMT2C*, *ARID1A*, and *CHD7* genes in our cohort. Our results suggest that mutations in epigenetic regulation genes may be valuable clinical markers and deserve further exploration in independent cohorts.

**Abstract:**

We have performed mutational profiling of 25 genes involved in epigenetic processes on 135 gastric cancer (GC) samples. In total, we identified 79 somatic mutations in 49/135 (36%) samples. The minority (*n* = 8) of mutations was identified in DNA methylation/demethylation genes, while the majority (*n* = 41), in histone modifier genes, among which mutations were most commonly found in *KMT2D* and *KMT2C*. Somatic mutations in *KMT2D*, *KMT2C*, *ARID1A* and *CHD7* were mutually exclusive (*p* = 0.038). Mutations in *ARID1A* were associated with distant metastases (*p* = 0.03). The overall survival of patients in the group with metastases and in the group with tumors with signet ring cells was significantly reduced in the presence of mutations in epigenetic regulation genes (*p* = 0.036 and *p* = 0.041, respectively). Separately, somatic mutations in chromatin remodeling genes correlate with low survival rate of patients without distant metastasis (*p* = 0.045) and in the presence of signet ring cells (*p* = 0.0014). Our results suggest that mutations in epigenetic regulation genes may be valuable clinical markers and deserve further exploration in independent cohorts.

## 1. Introduction

Gastric cancer (GC) is the 5th most common tumor in the world, and is the 3rd leading cause of cancer-related deaths worldwide. In 2018, more than 1,000,000 new GC patients were identified [1].

Recently, knowledge about the molecular mechanisms of gastric carcinogenesis has been intensively expanded. By using genome-wide approaches, The Cancer Genome Atlas (TCGA) Research Network divided GC into four molecular subtypes: Epstein-Barr associated (EBV), microsatellite instability (MSI), genomically stable (GS), and chromosomal instable (CIN) [2]. Next-generation sequencing (NGS) technologies have allowed identification of genes with an increased frequency of somatic mutations in different types of tumors. Those are the driver genes of carcinogenesis. Being used as targets for a therapy, such genes allow effective treatment of patients. However, GC is not enriched with mutations in known driver genes. Therefore, the targeted drugs that are useful in the treatment of other types of tumors are not effective in GC. Despite the intensive search for new drugs for cancer therapy, only trastuzumab and ramucirumab targeting HER2 and VEGFR2, respectively, are currently approved for GC treatment. Therefore, the search for novel genes with an increased somatic mutation frequency in GC is urgent to identify new clinical and prognostic markers, as well as new targets for treatment.

Epigenetic mechanisms, including DNA methylation/demethylation, covalent modifications of histone proteins (methylation, adenylation, phosphorylation, etc.), chromatin remodeling, and the action of non-coding RNAs create stable and clear patterns of gene expression during cell life. Epigenetic mechanism deregulations are associated with carcinogenesis, relapse of the disease, and metastasis, and can also serve as a useful clinical marker and a marker of response to therapy [3]. Application of NGS allowed identification of tumors without mutations in the known cancer driver genes that are, however, characterized by mutations in genes encoding epigenetic factors and chromatin-modifying enzymes. Today, deregulation of epigenetic mechanisms in different types of tumors has been confirmed, but its causes are insufficiently studied [4,5].

Somatic mutation profiling of epigenetic regulation genes will help to identify causes of epigenetic deregulation in GC and to suggest potential targets for successful therapy.

Using an NGS panel of 25 genes (*DNMT1, MBD1, TET1, DNMT3A, DNMT3B, EZH2, KDM6A, EP300, JARID1B, CREBBP, HDAC2, SIRT1, SMARCB1, SMARCA2, SMARCA4, ARID1A, ARID2, BRD7, PBRM1, CHD5, CHD7, CHD4, KMT2A, KMT2D and KMT2C*), we performed somatic mutation profiling in 135 tumor samples obtained from patients with GC.

## 2. Materials and Methods

### 2.1. Patients and Tumor Samples

The study included 135 patients with locally advanced GC who were treated in N.N. Burdenko Faculty Surgery Clinic, I.M. Sechenov First Moscow State Medical University from 2007 to 2015. The study was conducted in accordance with the Declaration of Helsinki and was approved by the Institutional Ethics Committee of I.M. Sechenov First Moscow State Medical University. Written informed consent was obtained from each participant in this study. All patients underwent surgical treatment, and resected tumor samples, as well as non-malignant gastric mucosa samples, were used in the study. GC was confirmed in all patients by morphological examination of the surgical material. For TNM staging, ESMO Clinical Practice Guidelines for diagnosis, treatment, and follow-up for gastric cancer [6] were used. The distribution of patients in clinical groups is presented in Table 2.

### 2.2. Mutation Screening by NGS

A total of 5 to 7, 10 μm paraffin sections were manually dissected to ensure that each sample contained at least 70% of neoplastic cells. Genomic DNA was isolated from archived samples using a QIAamp DNA FFPE Tissue Kit (Qiagen, Hilden, Germany), as recommended by the manufacturer.

Deep sequencing was performed using the Ion Torrent platform (ThermoFisher, Waltham, MA, USA) following established protocol [7]. The protocol includes the preparation of libraries of genomic DNA fragments, clonal emulsion PCR, sequencing, and bioinformatic analysis of obtained results. DNA fragment libraries were prepared using Ion Ampliseq ultra-multiplex PCR technology.

An epigenetic regulation genes panel with 1376 primer pairs was designed to amplify all coding regions, noncoding regions of the terminal exons, and putative splice site gene regions for 25 human genes: *DNMT1, MBD1, TET1, DNMT3A, DNMT3B, EZH2, KDM6A, EP300, JARID1B, CREBBP, HDAC2, SIRT1, SMARCB1, SMARCA2, SMARCA4, ARID1A, ARID2, BRD7, PBRM1, CHD5, CHD7, CHD4, KMT2A, KMT2D* and *KMT2C*. The panel was designed by using the Ion Ampliseq Designer v. 7.03 (ThermoFisher, Waltham, MA, USA). The total length of human genome sequences covered by the panel was 250,900 bp. The panel reached 98.09% coverage by design; this applies to exons and 25 bp flanking intron sequences. The information of the panel is shown in Appendix A. The selection of epigenetic regulation genes for the panel was based on the estimation of the frequency of their somatic mutations in GC, obtained from the COSMIC database and from the literature. Genes reported to be mutated in >3.5% of GC samples were included in the panel.

Multiplex PCR and subsequent stages of the fragment library preparation were performed using an Ion AmpliSeq Library Kit 2.0 (ThermoFisher, Waltham, MA, USA), according to the manufacturer’s protocol. Aliquots from the prepared libraries were subjected to clonal amplification on microspheres in the emulsion on the Ion Chef Instrument (ThermoFisher, Waltham, MA, USA). Sequencing was performed on the Ion S5 genomic sequencer according to the manufacturer’s protocol (ThermoFisher, Waltham, MA, USA) with the targeted sequencing depth of 1000×. The results were analyzed with Torrent Suite software consisting of Base Caller (the primary analysis of the sequencing results); Torrent Mapping Alignment Program—TMAP (alignment of the sequences to the reference genome GRCh37/hg19); and Torrent Variant Caller (analysis of variations in nucleotide sequences) with the cut-off for variant allele frequency set at 0.1, and minimum read depth of the variant allele set at 5. Genetic variants were annotated with ANNOVAR software [8]. Visual data analysis, manual filtering of sequencing artifacts, and sequence alignment were performed using the Integrative Genomics Viewer (IGV) [9].

### 2.3. Sanger Sequencing

Sanger sequencing was performed in order to (1) validate mutations detected by NGS screening and (2) distinguish somatic vs. germline mutations. For the second purpose, DNA samples extracted from archived non-malignant gastric mucosa of the same patients were used. The direct sequencing of individual PCR products from primers that flank areas of specific mutations were performed on the automatic genetic analyzer ABI PRISM 3500 (ThermoFisher, Waltham, MA, USA) according to the manufacturer’s protocols.

### 2.4. Statistical Analysis

Samples were compared using Fisher’s exact test. For more than 3 groups comparison Chi-squared test was used. Overall survival probability (OS) was calculated by the Kaplan–Meier product-limit method from the date of surgery till death by any cause and compared statistically using Mantel–Haenszel (log-rank) test. A groupwise mutual exclusivity test was carried out using the DISCOVER (Discrete Independence Statistic Controlling for Observations with Varying Rates) method, which is based on overall tumor-specific alteration rates to decide if alterations co-occur more or less than expected by chance and preventing spurious associations in co-occurrence detection with increasing statistical power to detect mutual exclusivities [10]. All calculations were conducted using R version 3.6.3 [R Core Team (2020). R: A language and environment for statistical computing. R Foundation for Statistical Computing, Vienna, Austria. URL https://www.R-project.org/ accessed on 7 August 2021].

### 2.5. Pathogenicity Prediction for Novel Mutations

To predict the pathogenicity of identified novel missense variants, a combination of PolyPhen2, PROVEAN, SIFT, and MutPred2 tools was used. I-Mutant 3.0 software was used to calculate the stability of the mutant protein. Loss of protein function effects were assessed with MutPred-LOF software. The effect of nonsynonymous substitutions on the structure was illustrated using the Project HOPE3D portal.

## 3. Results

### 3.1. The Spectrum of Detected Somatic Mutations

Using a targeted NGS panel for 25 epigenetic regulation genes, we performed mutational profiling in 135 tumor samples obtained from patients with GC. Our panel included the *DNMT1, MBD1, TET1, DNMT3A, DNMT3B* genes that control DNA methylation/demethylation; the *EZH2, UTX, EP300, JARID1B, CREBBP, HDAC2, SIRT1, KMT2A, KMT2D,* and *KMT2C* genes encoding histone modifiers; and the *SMARCB1, SMARCA2, SMARCA4, ARID1A, ARID2, BRD7, PBRM1, CHD5, CHD7, CHD4* genes responsible for chromatin remodeling. Mapped data depth and coverage for each sample are presented in Appendix A. For the analysis, we selected missense substitutions that were not annotated in the ClinVar, COSMIC, dbSNP databases and/or substitutions with a population frequency of MAF < 0.0005, as well as nonsense mutations and frameshift mutations. A total of 79 different mutations found in our cohort fulfilled the selection criteria. The variant allele frequency, total read depth, reference, and variant allele read depths, etc., for each of these mutations, are presented in Appendix A. No appropriate mutations were found in the *DNMT1, DNMT3A, EZH2, UTX, SMARCB1* and *SIRT1* genes. The identified mutations and their characteristics are presented in Table 1.

In total, we revealed 79 somatic mutations that fulfilled the selection criteria in 49/135 (36%) samples, and no mutations were found in the remaining samples. Among the identified variants, 29/79 were not annotated in dbSNP, 32/79 were not mentioned in gnomAD Exomes, and 68/79 were not mentioned in ClinVar.

The largest number of mutations was determined in histone modifier genes (41), and in chromatin remodeling genes (37). The smallest number was in DNA methylation/demethylation genes (8). Taking into consideration variation in the gene size, we normalized the mutation numbers in these three groups. Of the genes under study, histone modifier genes contained collectively 24,207 codons; chromatin remodeling genes, 16,891 codons, and DNA methylation/demethylation genes, 6188 codons. Thus, frequencies of mutations in these three groups were 0.0017, 0.0022, and 0.0013 per codon, respectively. These figures support a somewhat lower somatic mutation burden on the DNA methylation/demethylation genes, although the differences were not statistically significant. The distribution of variants in the epigenetic regulation genes in our patient samples was as follows: *KMT2D*-16, *ARID1A*-9, *KMT2A*-8, *KMT2C*-8, *CHD7*-7, *CHD5*-5, *CHD4, CREBBP*-4 each, *ARID2*, *SMARCA2, SMARCA4, DNMT3B* and *TET1*-3 each, *HDAC2, EP300, BRD7* and *MBD1*-2 each, and *PBRM1, JARID1B*-1. In 23/49 samples, a combination of more than one mutation in different genes was demonstrated, but mutations in *KMT2D, KMT2C, ARID1A,* and *CHD7* were significantly rarely found in one and the same sample (*p* = 0.038).

### 3.2. Pathogenicity Analysis of the Detected Mutations by Prediction Programs

For all novel mutations that fulfilled the selection criteria, pathogenicity analysis was performed by using prediction programs. By in silico analysis of pathogenicity for somatic alterations, we determined that 15/63 alterations were pathogenic according to more than two prediction tools. PolyPhen2-HumDiv predicted 26 of those as ‘Probably damaging’, and the other 15 were ‘Possibly Damaging’, whereas PolyPhen2-HumVar predicted 17 alterations as ‘Probably damaging’, 11 alterations as ‘Possibly damaging’, while other 35 alterations were ‘Benign’. However, it should be noticed that PolyPhen2-HumVar is more effective in mutations pathogenicity prediction for Mendelian disorders. 26/63 somatic alterations were predicted as ‘Deleterious’ by PROVEAN prediction tool; 42/63 variants were indicated as ‘Damaging’ by SIFT.

MutPred2 and MutPred-LOF are machine learning approaches, which incorporate genetic and molecular information to predict whether the alteration is pathogenic or not. We assigned a threshold value of 0.68 for pathogenic, as recommended by developers, because it yields a false positive rate of 10%. With this assumption, 11/63 somatic missense variants were predicted as pathogenic, as well as and 10/16 nonsense and frameshift variants, by MutPred-LOF with a cut-off value of 0.50 (as recommended for MutPred-LOF).

I-Mutant 3.0 predicts protein stability changes based on a protein sequence or protein structure by using a support vector machine training algorithm. The I-Mutant 3.0 predicted a decrease in protein structure stability for 44 somatic alterations and an increase for the other 19 (Appendix A).

### 3.3. Analysis of Clinical Significance of Mutations in Epigenetic Regulation Genes

The distribution of mutations in our patient cohort aligned to clinical features is shown in Figure 1.

We found no associations of overall somatic mutation status (absence of mutations vs. presence of at least one mutation) of epigenetic regulation genes with gender, age, tumor size, lymph node metastases, stage, anatomical localization, Lauren type, distant metastases, and presence of signet ring cells (Table 2). As for individual genes, we have only discovered that mutations in *ARID1A* were associated with distant metastases (*p* = 0.03).

In the analysis of survival using the Kaplan–Meier method, we found that the overall survival of patients in the group with metastases and the group of tumors with signet ring cells was significantly reduced in the presence of mutations in the epigenetic regulation genes (*p* = 0.036 and *p* = 0.042, respectively) comparing with patients without mutations (Figure 2).

Somatic mutations in the chromatin remodeling genes correlate with a low survival rate of patients in the absence of distant metastases (*p* = 0.045) and with the presence of signet ring cells in tumors (*p* = 0.0014) (Figure 3).

For the group of histone-modifying genes, no significant clinical correlations were found. The group with mutations in the DNA methylation/demethylation genes included only 8 patients and was too small to perform statistical analysis.

## 4. Discussion

Somatic mutations in epigenetic regulation genes are not very common in GC and were determined only in 36% samples (49/135) in our study. Mutations were most rarely detected in genes regulating DNA methylation/demethylation. We have not found any somatic mutations in *DNMT1* and *DNMT3A*. Besides, the group of patients with mutations in the DNA methylation-related genes (*MBD1*, *TET1*, *DNMT3B*) was the smallest one with only 8 out of 135 patients. Such a low frequency may be a result of the cancer type being investigated. Chai-Jin Lee et al. demonstrated that frequencies of somatic mutations in genes associated with DNA methylation and demethylation (*DNMT1*, *DNMT3A*, *MBD1*, *MBD4*, *TET1*, *TET2* and *TET3*) significantly varied in different types of cancers. Thus, in myeloid leukemia samples, the frequency of *DNMT1* and *DNMT3A* mutations was high, whereas, in glioblastoma, renal cell carcinoma, and colon carcinoma, the total mutation rate was less than 9% [11]. The low frequency of mutations in the DNA methylation drivers in solid tumors is consistent with our results. Many studies have been published demonstrating DNA methylation as a clinical marker of carcinogenesis; however, the role of somatic mutations in genes regulating methylation/demethylation in solid tumors has not yet been sufficiently investigated. Moreover, although we did not find any *DNMT3A* mutations in our samples, they were identified in other solid tumors. In 1.2% of papillary thyroid carcinoma cases, mutations and/or loss of *DNMT3A* expression were associated with aggressive clinical course and poor outcome [12].

In our work, the largest number of mutations was detected in histone modification genes (52%, 41/79), with 16 mutations in *KMT2D*, 8 in *KMT2C,* and 8 in *KMT2A*. The proteins encoded by these KMT2 (histone-lysine N-methyltransferases subclass 2) genes were components of a COMPASS-like complex that performs mono-, di-, and trimethylation of lysine 4 (H3K4) in histone 3 and is associated with transcription activation, facilitating access of transcription factors to the promoter and enhancer regions of genes [13]. The functions of COMPASS complexes are vitally important for the normal development of an organism, and mutations in genes encoding their protein components are associated with carcinogenesis [14]. KMT2C and KMT2D proteins restrain cell proliferation and could be considered tumor suppressors [15]. In addition to lysine methylation associated with transcription activation, methyltransferases KMT2C and KMT2D play an important role in the maintenance of genomic stability and DNA repair [16]. Besides, these proteins, together with PTIP (PAX transactivation-domain interacting protein), a subunit of the KMT2C/KMT2D complexes, were found to increase the instability and induce the degradation of the MRE11-dependent replication fork in BRCA-deficient cells [17].

The *KMT2D* and *KMT2C* genes are among the most frequently mutated in cancers, which is also confirmed by our study. Mutations were detected in various types of solid tumors, such as melanoma, urothelial carcinoma, lung cancer, as well as in esophageal and stomach cancers [18].

In our study, *KMT2D* mutations had the highest frequency of 12% and were distributed throughout the gene (Figure 4). Mutations of the *KMT2D* gene are mainly localized in the central part of the gene coding sequence, which corresponds to the protein region between the PHD-finger domain and the SET domain. This is also in concordance with the data obtained by other authors [19].

According to the analysis by pathogenicity prediction programs, one of the novel somatic missense mutations that we identified in the *KMT2D* gene, p.R3727C, was determined as pathogenic by almost all prediction tools. This substitution results in disruption of the leucine zipper motif, which was necessary for the protein–protein interactions or dimerization [20]. Disruption of the leucine zipper motif seriously alters the function of proteins, which leads to a deregulation of protein interactions and blocking transcription. Directed alterations of the leucine zipper motif are currently created in synthetic proteins that are used as antitumor drugs [21].

The analysis of pathogenicity of unannotated mutations identified by us in *KMT2C* revealed three mutations (p.R973G, p.M959I, p.C1953Y) that were pathogenic according to three or more prediction tools. The first two of them were located in the PHD-finger domain of the gene, and p.C1953Y was located in the disorder domain. Disorder domains are characterized by high instability, and substitutions in this region can change the protein conformation. Recent studies have demonstrated that around 20% of mutations in cancers are located in these regions, causing abnormalities of protein conformations and functions [22]. Mutations in *KMT2C* in diffuse GC are associated with epithelial–mesenchymal transition (EMT) and acquisition of the mesenchymal phenotype by cells and are also markers of a poor prognosis [23]. Mutation distribution along the *KMT2C* gene is shown in Figure 5.

In our study, mutations in the *KMT2D* and *KMT2C* were significantly rarely combined in one sample (*p* = 0.038). There is a hypothesis that mutually exclusive genomic events are functionally related by common biological pathways, and mutually exclusive genes act on the same downstream effectors, thereby demonstrating functional redundancy. Therefore, the aberration of one of these genes is enough to completely disrupt their common pathways [24]. The *KMT2D* and *KMT2C* are components of similar COMPASS complexes that perform the same function. Deregulation of either *KMT2C* or *KMT2D* separately can serve as a driver mutation at the early stages of carcinogenesis, leading to changes in the epigenomic landscape. As was demonstrated for bladder cancer, tumor cells with low KMT2C activity experienced a deficiency of DNA repair mediated by homologous recombination and suffer from endogenous DNA damage and genomic instability, and their treatment with the PARP1/2 inhibitor olaparib leads to synthetic lethality [16]. The high frequency of *KMT2D* and *KMT2C* mutations in GC and its associations with repair processes allows considering them as targets for tumor treatment using PARP inhibitors, causing the lethality of tumor cells.

We compared our result on mutual exclusivity of *KMT2D* and *KMT2C* mutations with other GC mutation databases. Three datasets were acquired using cBioPortal (http://cbioportal.org accessed on 7 August 2021): Gastric Cancer (OncoSG, 2018), Stomach Adenocarcinoma (Pfizer and UHK), and TCGA PanCancer Stomach Adenocarcinoma (STAD). Visual analysis suggested that *KMT2D* and *KMT2C* mutations in these datasets were not mutually exclusive (Figure 6). For statistical analysis, we retained only sequenced samples with mutation data (without Copy-Number Alterations) in all three studies. For the groupwise mutual exclusive test, *p*-values were as follows: 0.088 for Onco SG, 0.016 for TCGA STAD, and 0.5 for the Pfizer study. Using the wFisher *p*-value combination method [25] with sample size for each experiment, we obtained the *p*-value of the mutual exclusive test under the nominal significance level of 0.05 (Figure 6a). Another interesting observation was that considering missense mutations only, mutations in *KMT2D* and *KMT2C* visually were almost mutually exclusive in these three datasets, as they were in our study (Figure 6b), although calculated differences did not approach a significance level of 0.05, which we attributed to the sample sizes. In this respect, we paid attention to the studies of bigger sample size, though of another cancer localization, namely, Breast Cancer METABRIC, Nature 2012, and Nat Commun 2016 (2509 samples) and Breast Cancer MSK, Cancer Cell 2018 (1918 samples), and in these datasets, we witnessed obvious mutual exclusivity of somatic mutations in *KMT2D*, *KMT2C,* and *ARID1A*. Although this may be a cancer type-specific observation, we altogether cannot rule out sample size effect and/or peculiarities of mutation detection/interpretation in different studies.

The *ARID1A* and *CHD7* genes that are related to chromatin remodeling were often mutated in our patient samples. The *ARID1A* is often mutated in esophageal and gastric cancers and is the canonical cancer gene according to the Cosmic Cancer Gene Census [26]. The proteins encoded by the *ARID1A*, *SMARCA1*, *SMARCA2*, and *SMARCA4* are subunits of the conservative multisubunit SWI/SNF complex, which uses the energy of ATP hydrolysis to mobilize nucleosomes and remodel chromatin. The expression of these genes is often deregulated in the esophagus and gastric cancers [27].

ARID1A substitutions that we identified in gastric tumors, not annotated in human mutation databases, namely p.R2236C, p.Q1415H, and p.P1710L, are of interest since they can lead to deregulation of molecular mechanisms important for cancer progression. According to the results of in silico analysis, the p.R2236C substitution results in aberration of ADP-ribosylation, which is important for the DNA damage repair, as well as for the formation of an allosteric site at p.R2233 that can be used to bind therapeutic agents. Today, the search and targeting of allosteric sites are one of the strategies in the development of antitumor drugs [28].

ARID1A:p.Q1415H and ARID1A:p.P1710L amino acid substitutions demonstrate an overall predicted loss of O- and C-linked glycosylation. Post-translational modifications, such as glycosylation, affect the transport, stability, and folding of the protein, changing its biochemical and biophysical properties. Numerous studies confirmed that changes in protein glycosylation have a great impact on carcinogenesis and contribute to the appearance of more aggressive cell phenotypes [29].

Recent studies demonstrated that mutations in genes and abnormal expression of the ISO/SNF complex proteins that participate in chromatin remodeling were associated with a more aggressive course of the disease, as well as with EBV and MSI subtypes of GC [30]. In our study, somatic mutations in the chromatin remodeling genes were also found to be associated with worse overall survival in patients (without distant metastases, *p* = 0.045; and in the presence of signet ring cells, an indicator of the aggressive course in GC, *p* = 0.00011). H. Takeshima et al. investigated the role of chromatin remodelers in GC and suggested that deregulations of chromatin remodeling occur at an early stage of gastric carcinogenesis and are involved in the formation of the field cancerization [31].

Investigation of GC using NGS previously revealed that 47% of gastric adenocarcinomas were characterized by mutations of chromatin remodeling genes, and somatic mutations of *ARID1A* had a high frequency, as it was in our study. It was shown that gastrointestinal tumors with *ARID1A* mutations demonstrated high immune activity [32]. Gastric carcinomas with somatic *ARID1A* mutations were characterized by a more intense PD-L1 expression than tumors without mutations. PD-L1 overexpression contributes to a more active response to immunotherapy and a better prognosis of survival for patients with mutations in *ARID1A* compared to tumors with wild-type *ARID1A*. *ARID1A* mutations can serve as a biomarker for the identification of patients with gastrointestinal cancer who are sensitive to immunotherapy [33]. Clinically, the loss of *ARID1A* expression was correlated with larger tumor size, deeper invasion, lymph node metastasis, and a poor prognosis [34]. In line with these observations, in our study, mutations in *ARID1A* were associated with distant metastases (*p* = 0.03).

## 5. Conclusions

As a result of somatic mutation profiling of epigenetic regulation genes in GC, we have revealed associations of the presence of such mutations in tumors with a decrease in patient survival and the risk of developing distant metastasis, making the presence of mutations a marker of a poor prognosis. Studying mutations in epigenetic regulation genes can also contribute to the development of new approaches to drug therapy for GC treatment, adding to them PARP inhibitors for the treatment of tumors with mutations in genes of the *KMT2* family and immunotherapy for the treatment of tumors with *ARID1A* mutations. According to our results, this may be a significant group of patients, as the total frequency of mutations in the chromatin remodeling genes and histone modifiers in our sample were approximately 25% of all patients with mutations in epigenetic regulation genes.

## Figures and Tables

**Figure 1 cancers-13-04586-f001:**
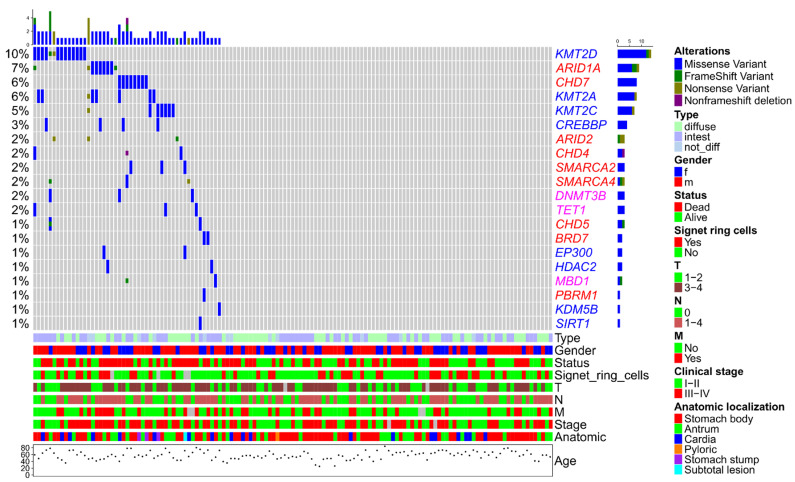
Spectrum of epigenetic regulation genes somatic mutations in gastric cancer. Gene names are marked according to functions of the encoded proteins: histone modifiers, blue; chromatin remodeling, red; DNA methylation/demethylation, magenta.

**Figure 2 cancers-13-04586-f002:**
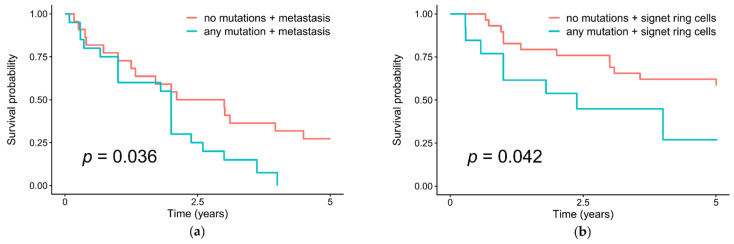
Overall survival in patients with and without somatic mutations of the epigenetic regulation genes in their gastric tumors, and (**a**) with distant metastases; (**b**) with the presence of signet ring cells in tumors.

**Figure 3 cancers-13-04586-f003:**
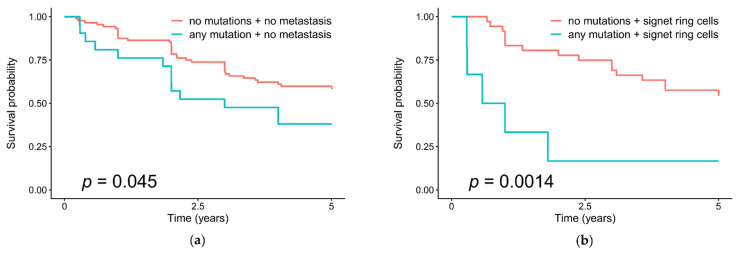
Overall survival in patients with and without somatic mutations of the chromatin remodeling genes in their gastric tumors, and (**a**) with distant metastases; (**b**) with the presence of signet ring cells in tumors.

**Figure 4 cancers-13-04586-f004:**
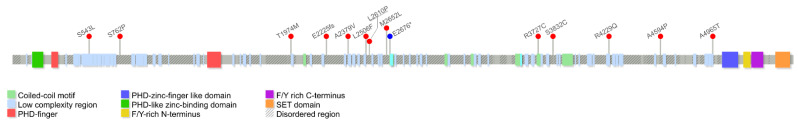
Distribution of the *KMT2D* mutations detected in this study, along the gene.

**Figure 5 cancers-13-04586-f005:**
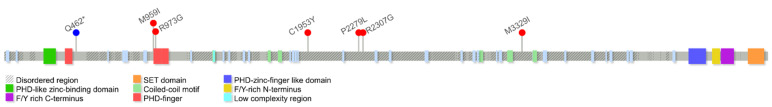
Distribution of the *KMT2C* mutations detected in this study, along the gene.

**Figure 6 cancers-13-04586-f006:**
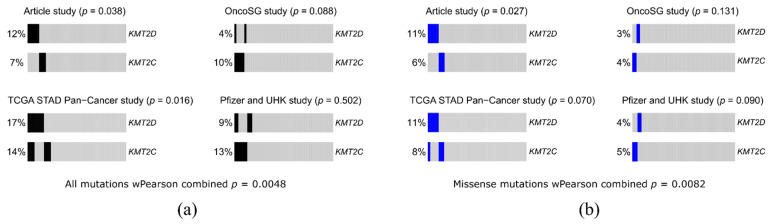
Analysis of mutual exclusivity of *KMT2D* and *KMT2C* mutations on the data presented in this article and in other gastric cancer mutation databases. Portions of samples without mutations in *KMT2D* or *KMT2C* are shown in grey; (**a**) analysis of all types of mutations, excluding amplification and deep deletions, portions of samples with mutations in *KMT2D* or *KMT2C* are colored black; (**b**) analysis of missense mutations only, portions of samples with missense mutations in *KMT2D* or *KMT2C* are colored blue.

**Table 1 cancers-13-04586-t001:** Somatic mutations detected in epigenetic regulation genes in 135 gastric tumors.

№	Gene/Mutation	Position According to hg19	rsID	MAF (gnomAD Exomes)	ClinVar	# of Cases
1	ARID1A:exon1:c.G544A:p.A182T	chr1:27023438	-	-	-	1
2	ARID1A:exon16:c.G3902A:p.S1301N	chr1:27100106	rs989613588	A = 4 × 10^−6^	Not Reported	1
3	ARID1A:exon18:c.G4245C:p.Q1415H	chr1:27100963	-	-	Not Reported	1
4	ARID1A:exon20:c.C6706T:p.R2236C	chr1:27107095	rs763691986	T = 2 × 10^−5^	Not Reported	1
5	ARID1A:exon20:c.C5483G:p.S1828X	chr1:27105872	-	-	Not Reported	1
6	ARID1A:exon20:c.C5129T:p.P1710L	chr1:27105518	-	-	Not Reported	1
7	ARID1A:exon2:c.G1330A:p.G444S	chr1:27056334	rs541301347	A = 2.4 × 10^−5^	Not Reported	1
8	ARID1A:exon20:c.5881_5887del:p.S1961fs	chr1:27106270	-	-	Not Reported	1
9	ARID1A:exon18:c.4713_4714del:p.N1571fs	chr1:27101431	-	-	Not Reported	1
10	ARID2:exon1:c.53_57del:p.A18fs	chr12:46123672	-	-	Not Reported	1
11	ARID2:exon8:c.C820T:p.R274X	chr12:46230571	-	-	Likely pathogenic	1
12	ARID2:exon8:c.C985T:p.Q329X	chr12:46230736	-	-	Not Reported	1
13	SMARCA2:exon7:c.C734T:p.T245M	chr9:2192726	rs753433101	A = 3 × 10^−5^	Not Reported	1
14	SMARCA2:exon7:c.A1256G:p.K419R	chr9:2056754	-	-	Not Reported	1
15	SMARCA2:exon7:c.G1202A:p.R401H	chr9:2056700	rs745500947	T = 8 × 10^−6^	Not Reported	1
16	SMARCA4:exon18:c.C2738T:p.P913L	chr19:11132522	rs778175819	T = 4 × 10^−6^	Not Reported	1
17	SMARCA4:exon3:c.C430T:p.Q144X	chr19:11096939	-	-	Not Reported	1
18	SMARCA4:exon3:c.583delC:p.P195fs	chr19:11097092	-	-	Not Reported	1
19	KDM5B:exon17:c.C2392T:p.R798W	chr1:202711581	rs1189771603	A = 2.8 × 10^−5^	Not Reported	1
20	CHD4:exon28:c.C4216T:p.R1406C	chr12:6692034	-	-	Not Reported	1
21	CHD4:exon4:c.417_419del:p.139_140del	chr12:6711145	rs71584865	del(TCC)3 = 8 × 10^−6^	Not Reported	2
22	CHD4:exon4:c.C421G:p.P141A	chr12:6710929	-	-	Not Reported	1
23	CHD5:exon2:c.119delT:p.F40fs	chr1:6228298	-	-	Not Reported	1
24	CHD5:exon15:c.C2257A:p.L753M	chr1:6202367	-	-	Not Reported	1
25	CHD5:exon7:c.G910A:p.A304T	chr1:6211176	rs768430028	T = 1.3 × 10^−4^	Not Reported	2
26	CHD5:exon2:c.A156T:p.K52N	chr1:6228261	rs964095593	A = 3 × 10^−5^	Not Reported	1
27	CHD7:exon31:c.G6112A:p.D2038N	chr8:61765396	rs747846723	A = 6 × 10^−5^	Uncertain Significance	1
28	CHD7:exon35:c.A7819G:p.S2607G	chr8:61773673	rs1424434796	G = 5 × 10^−6^	Not Reported	2
29	CHD7:exon22:c.G4859A:p.R1620Q	chr8:61757431	rs768497646	A = 2.9 × 10^−5^	Not Reported	1
30	CHD7:exon1:c.G749A:p.R250H	chr8:61654740	rs767475667	A = 6 × 10^−5^	Not Reported	1
31	CHD7:exon22:c.G5017A:p.D1673N	chr8:61757589	rs769563309	A = 2.4 × 10^−5^	Not Reported	1
32	CHD7:exon29:c.G5828A:p.R1943Q	chr8:61764740	rs753723769	A = 2.8 × 10^−5^	Uncertain Significance	1
33	EP300:exon3:c.A752G:p.N251S	chr22:41521890	rs142009367	G = 2.2 × 10^−4^	Benign	2
34	HDAC2:exon13:c.C1430A:p.T477N	chr6:114262878	rs1341257540	-	Not Reported	1
35	HDAC2:exon6:c.G511A:p.V171I	chr6:114274569	-	-	Not Reported	1
36	CREBBP:exon30:c.C6335T:p.P2112L	chr16:3778599	rs587783512	A = 1.6 × 10^−5^	Uncertain Significance	1
37	CREBBP:exon2:c.C458T:p.P153L	chr16:3900638	rs146538907	A = 3.35 × 10^−4^	Likely Benign	1
38	CREBBP:exon3:c.C922T:p.P308S	chr16:3860657	-	-	Not Reported	1
39	CREBBP:exon18:c.A3421C:p.K1141Q	chr16:3807998	-	-	Not Reported	1
40	BRD7:exon7:c.A871G:p.S291G	chr16:50368638	rs200218240	C = 1.6 × 10^−4^	Not Reported	1
41	BRD7:exon5:c.C571T:p.Q191X	chr16:50383954	-	-	Not Reported	1
42	PBRM1:exon17:c.C2032T:p.R678C	chr3:52643864	rs1422119249	-	Not Reported	1
43	KMT2A:exon27:c.T9737C:p.I3246T	chr11:118376344	rs1259638674	C = 3 × 10^−5^	Not Reported	1
44	KMT2A:exon27:c.C9947T:p.A3316V	chr11:118376554	rs201447376	T = 1.3 × 10^−4^	Not Reported	1
45	KMT2A:exon27:c.G10181A:p.G3394E	chr11:118376788	rs782460936	A = 3 × 10^−5^	Not Reported	1
46	KMT2A:exon21:c.G5726A:p.W1909X	chr11:118368712	-	-	Not Reported	1
47	KMT2A:exon27:c.G9247A:p.V3083I	chr11:118375854	-	-	Not Reported	1
48	KMT2A:exon30:c.A10984G:p.S3662G	chr11:118380746	rs201724738	G = 5.6 × 10^−5^	Not Reported	1
49	KMT2A:exon36:c.G11903A:p.R3968Q	chr11:118392871	rs369182428	A = 3 × 10^−5^	Not Reported	1
50	KMT2A:exon30:c.A10975G:p.S3659G	chr11:118380746	rs201724738	G = 1 × 10^−4^	Not Reported	1
51	KMT2C:exon43:c.G9987A:p.M3329I	chr7:151860675	rs200804156	T = 6 × 10^−5^	Not Reported	1
52	KMT2D:exon10:c.T2284C:p.S762P	chr12:49445182	-	-	Not Reported	1
53	KMT2C:exon10:c.C1384T:p.Q462X	chr7:151949716	-	-	Not Reported	1
54	KMT2C:exon36:c.C6836T:p.P2279L	chr7:151878109	rs150844259	A = 6 × 10^−5^	Not Reported	1
55	KMT2C:exon36:c.A6919G:p.R2307G	chr7:151878026	rs772283102	C = 2.4 × 10^−5^	Not Reported	1
56	KMT2C:exon36:c.G5858A:p.C1953Y	chr7:151879087	-	-	Not Reported	1
57	KMT2C:exon18:c.A2917G:p.R973G	chr7:151927067	rs60244562	-	Not Reported	2
58	KMT2C:exon18:c.G2877A:p.M959I	chr7:151927107	rs4024402	-	Not Reported	1
59	KMT2D:exon31:c.A7954C:p.M2652L	chr12:49433599	rs147706410	G = 4.5 × 10^−4^	Likely Benign	1
60	KMT2D:exon41:c.G13780C:p.A4594P	chr12:49424443	rs545972414	G = 2.5 × 10^−4^	Not Reported	2
61	KMT2D:exon28:c.C5921T:p.T1974M	chr12:49436060	rs777415982	A = 6 × 10^−5^	Not Reported	1
62	KMT2D:exon48:c.G14893A:p.A4965T	chr12:49420856	rs200747934	T = 1.53 × 10^−4^	Uncertain Significance	2
63	KMT2D:exon39:c.G12686A:p.R4229Q	chr12:49445262	rs753607446	T = 3.2 × 10^−5^	Not Reported	1
64	KMT2D:exon31:c.T7829C:p.L2610P	chr12:49433724	rs200998047	G = 1.85 × 10^−4^	Uncertain Significance	1
65	KMT2D:exon31:c.6673delG:p.E2225fs	chr12:49434880	-	-	Not Reported	1
66	KMT2D:exon31:c.C7516T:p.L2506F	chr12:49434037	rs749670394	A = 4 × 10^−6^	Not Reported	1
67	KMT2D:exon39:c.C11179T:p.R3727C	chr12:49427309	rs566069597	A = 3.6 × 10^−5^	Not Reported	1
68	KMT2D:exon10:c.C1628T:p.S543L	chr12:49445838	rs776242478	A = 8 × 10^−6^	Not Reported	1
69	KMT2D:exon31:c.G8026T:p.E2676X	chr12:49433527	-	-	Not Reported	1
70	KMT2D:exon31:c.C7136T:p.A2379V	chr12:49434417	rs200842315	A = 6 × 10^−5^	Likely benign	1
71	KMT2D:exon39:c.C11495G:p.S3832C	chr12:49426993	-	-	Not Reported	1
72	MBD1:exon9:c.G796A:p.E266K	chr18:47801382	rs142015383	T = 5 × 10^−4^	Not Reported	1
73	MBD1:exon8:c.734delC:p.P245fs	chr18:47801527	rs1173827934	-	Not Reported	1
74	TET1:exon4:c.G2407A:p.A803T	chr10:70404893	rs765094207	A = 2.4 × 10^−5^	Not Reported	1
75	TET1:exon2:c.G320A:p.R107Q	chr10:70332415	rs1419371452	A = 8 × 10^−6^	Not Reported	1
76	TET1:exon4:c.G3476A:p.R1159Q	chr10:70405962	rs140289196	A = 2.2 × 10^−4^	Not Reported	1
77	DNMT3B:exon19:c.G2138A:p.R713Q	chr20:31390243	rs747182299	A = 3 × 10^−5^	Likely Pathogenic	1
78	DNMT3B:exon6:c.A680G:p.Y227C	chr20:31379501	-	-	Not Reported	1
79	DNMT3B:exon17:c.G1855A:p.E619K	chr20:31388054	rs576798456	A = 8 × 10^−6^	Not Reported	1

**Table 2 cancers-13-04586-t002:** Clinical characteristics of patients and their distribution by groups with mutations (mut+) and without mutations (mut−) in epigenetic regulation genes.

Parameters		Total Cases	Mut−	Mut+	*p*-Value *
		135	86	49	
Age	<50	35	22	13	1
	>50	100	64	36	
Sex	m	83	51	32	0.58
	f	52	35	17	
Survival status	Alive	59	42	17	0.14
	Dead	76	44	32	
5-year survival status	Alive	60	42	18	0.19
	Dead	69	40	29	
	<5 years follow-up	6	4	2	
T	T1-2	49	33	16	0.46
	T3-4	84	51	33	
	is	2	2	-	
N	N0	56	37	19	0.71
	N1-3	79	49	30	
M	M0	89	59	30	0.55
	M1	42	25	17	
	Unknown	4	-	-	
Lauren classification	Diffuse	59	41	18	0.26
	Intestinal	64	38	26	
	Not Differentiate	12	7	5	
Stage	I-II	57	39	18	0.36
	III-IV	76	45	31	
	Unknown	2	-	-	
Anatomical localization	Stomach body	75	52	23	0.10
	Antrum	36	22	14	
	Cardia	20	11	8	
	Stomach stump	3	0	3	
	Subtotal lesion	1	0	1	
	Pyloric	1	1	0	
Signet ring cells	yes	42	29	13	0.69
	no	89	57	32	
	Unknown	4	0	4	

* patients group with mutations in epigenetic regulation genes (mut+) vs. without mutations (mut−).

## Data Availability

The data presented in this study are available in the article and Appendix A.

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
