# Peer review of "Mutations in Epigenetic Regulation Genes in Gastric Cancer"

_cancers, 2021, doi:10.3390/cancers13184586_

Round 1

Reviewer 1 Report

In this manuscript, Nemtsova et al analyzed genetic mutations within 25 genes which regulate epigenetics in a large cohort of patients with gastric cancer. The analysis of 135 GC patients found that the majority of somatic mutations are observed in histone modifier genes; while DNA-methylation/demethylation genes have a low rate of mutations. The highest level of mutations was found in KMT2D and KMT2C genes. The authors show that the identified mutations strongly correlate with low survival rate of GC patients.  The authors further present a detailed analysis of consequences of the mutations for the biological functions of proteins. This is a very interesting and highly significant work. However, there are several minor issues that need to be addressed by the authors.

Comments:

  • The list of examined genes misses HDAC1 which is an important member of histone deacetylase family proteins. It is not clear why this gene was not included in the analysis.
  • Figure 5 shows distribution of mutations within the KMT2D gene, which is very useful. R3727A mutation is particularly interesting since it disrupts leucine zipper structure and protein-protein interactions. Since three mutations in KMT2C gene are pathogenic, a similar distribution map should be included for mutations within the KMT2C gene.
  • It appears that the identified mutations resulted in substitution of amino acids. Did the author find mutations that create stop codons?    
  • In several sections, the authors state that the identified mutations are the cause of pathological changes. These statements should be corrected since cause-effect relationships were not addressed in this manuscript.  For example in Simple Summary, the authors state that “Mutations in DNA-methylation/demethylation genes reduce the survival rate…..”. I would suggest to change this to “correlate with low survival rate”.

Author Response

Dear reviewer,

Thank you for your positive estimation of our work.

We have addressed your comments as follows:

  • The list of examined genes misses HDAC1 which is an important member of histone deacetylase family proteins. It is not clear why this gene was not included in the analysis.

Selection of epigenetic regulation genes for the panel was based on the estimation of frequency of their somatic mutations in GC, obtained from COSMIC database and from literature. Genes reported to be mutated in >3.5% of GC samples were included in the panel. We now indicate this in the “Materials and Methods” section. It looks like HDAC1 did not cross the threshold for somatic mutation frequency. We have now checked the Gastric Cancer (OncoSG, 2018), Stomach Adenocarcinoma (Pfizer and UHK, Nat Genet 2014), Stomach Adenocarcinoma (TCGA, PanCancer Atlas) databases, and indeed, HDAC1 is very rarely mutated in gastric cancer.

  • Figure 5 shows distribution of mutations within the KMT2D gene, which is very useful. R3727A mutation is particularly interesting since it disrupts leucine zipper structure and protein-protein interactions. Since three mutations in KMT2C gene are pathogenic, a similar distribution map should be included for mutations within the KMT2C gene.

We have included a figure for mutation distribution along the KMT2C gene.

  • It appears that the identified mutations resulted in substitution of amino acids. Did the author find mutations that create stop codons?

Yes, such mutations were also found. They are annotated as nonsense and frameshift variants on Figure 1, and listed in Table 1.

  • In several sections, the authors state that the identified mutations are the cause of pathological changes. These statements should be corrected since cause-effect relationships were not addressed in this manuscript.  For example in Simple Summary, the authors state that “Mutations in DNA-methylation/demethylation genes reduce the survival rate…..”. I would suggest to change this to “correlate with low survival rate”.

We have corrected this statement in Simple Summary and elsewhere in the text.

Reviewer 2 Report

In this manuscript, Nemtsova, et al. reported their study of 25 epigenetic regulator genes in 135 gastric cancer samples. They identified 79 somatic mutations in 49/135 samples, and the these mutations were found to be significantly associated with reduced overall survival of patients with metastases and patients with tumors with signet ring cells. Among these somatic mutations, the majority were found in histone modifier genes and the minority was identified in DNA methylation/demethylation genes.

Overall, the authors demonstrated a good dataset about the mutation profile of epigenetic regulator genes in gastric cancer, and their results suggest these genes may be a valuable clinical marker and can be further studied. However, there are still some issues that need to be fixed, and some improvement could be done before publishing on Cancers.

Major Issues:

  1. On page 7, 163-171/252-267: the authors claimed the largest number of mutations was in histone modifier genes, and the smallest was in DNA methylation/demethylation genes. However, in all 25 genes, there is a huge variation in the gene size. For example, KMT2D is 5537 AA, but HDAC2 is only 488 AA. Directly compare the mutation numbers is biased because larger genes could enrich more mutations. The authors should normalize the mutation number with the gene size and reconsider the conclusions.

  2. Based on the somatic mutation profile, the authors discussed several points about the mutation frequency, association, mutually exclusive, etc. However, this is still a relatively small cohort, and patients were from the single center. In addition, the information provided in the manuscript could not reflect the enrollment is unbiased. So, compare this cohort with other published datasets is essential. For example, the Gastric Cancer (OncoSG, 2018), Stomach Adenocarcinoma (Pfizer and UHK, Nat Genet 2014), Stomach Adenocarcinoma (TCGA, PanCancer Atlas), etc. By comparing these data, the authors may find out KMT2D and KMT2C are not mutually exclusive in these datasets, and there is a 50% overlap. The authors should do a comparative study and discuss about the difference.

  3. In the materials and methods section, the authors should provide the summary of sequencing and key-parameters in mutation calling, including sequencing depth and coverage in each sample, minimum depth and VAF cutoff for mutation sites, etc.

  4. The authors claimed the mutations they identified are somatic mutations, but they did not describe whether they used normal control to distinguish somatic mutations or just based on VAF. In addition, the variant allele frequency of all mutations is essential information, must be provided.

  5. In table 2, some of the numbers could not add up to the total number. For example, in section Signet ring cells, Mut+: 13+32=45, but the total number should be 49. And for total cases column, 42+89=131, not 135. The authors should check all the numbers to make sure all the results are correct. For section T/N/M, the authors should describe what are the T/N/M.

  6. Figure 3 and line 216-220. the group with mutations in the DNA methylation/demethylation genes only included 8 patients. The results from this data set is not helpful because of small patient group. Moreover, base on figure 3, all 8 patients were dead at the same time point? It doesn't make sense.

Minor Issues:

  1. In line 73-75, and in Figure 1, gene names should use the italic font.

  2. In Figure 1, the gene names can be marked with different colors to help distinguish the gene groups described in text.

  3. The authors used a designed panel that cover 25 genes, does this panel reach 100% coverage? Since the sequencing panel is an key material in this manuscript, the authors should provide the information of this panel in the supplementary.

  4. In table 1, column MAF: The numbers should be edited using scientific notation instead of using a lot of zeros. And there are two mutations MAF were marked as 0.000, is this a mistake or do the authors have a specific purpose? Because these two mutations doesn't have data in gnomAD.

  5. In the manuscript, the authors made several mistakes in the float number of the p-value (use "," as "."). For example, in line 211, 217, 218, 226, etc.

Author Response

Dear Reviewer,

Thank you very much for having carefully read our manuscript and for your valuable criticisms.

Major Issues:

  1. On page 7, 163-171/252-267: the authors claimed the largest number of mutations was in histone modifier genes, and the smallest was in DNA methylation/demethylation genes. However, in all 25 genes, there is a huge variation in the gene size. For example, KMT2D is 5537 AA, but HDAC2 is only 488 AA. Directly compare the mutation numbers is biased because larger genes could enrich more mutations. The authors should normalize the mutation number with the gene size and reconsider the conclusions.

We have addressed this issue and added the following to the text: “Taking into consideration variation in the gene size, we have normalized the mutation numbers in these three groups. Of the genes under study, histone modifier genes contain collectively 24207 codons; chromatin remodeling genes, 16891 codons, and DNA methylation/demethylation genes, 6188 codons. Thus, frequencies of mutations in these three groups are 0.0017, 0.0022 and 0.0013 per codon, respectively. These figures support a somewhat lower somatic mutation burden on the DNA methylation/demethylation genes, although the differences are not statistically significant”.

In Discussion, we substituted “…the most often mutated genes KMT2D (16/41), KMT2C (8/41), and KMT2A (8/41)” with “16 mutations in KMT2D, 8 in KMT2C, and 8 in KMT2A”, to avoid repetitive discussion.

  1. Based on the somatic mutation profile, the authors discussed several points about the mutation frequency, association, mutually exclusive, etc. However, this is still a relatively small cohort, and patients were from the single center. In addition, the information provided in the manuscript could not reflect the enrollment is unbiased. So, compare this cohort with other published datasets is essential. For example, the Gastric Cancer (OncoSG, 2018), Stomach Adenocarcinoma (Pfizer and UHK, Nat Genet 2014), Stomach Adenocarcinoma (TCGA, PanCancer Atlas), etc. By comparing these data, the authors may find out KMT2Dand KMT2C are not mutually exclusive in these datasets, and there is a 50% overlap. The authors should do a comparative study and discuss about the difference.

Our cohort (135 samples) is comparable in size with those of OncoSG, 2018 (147 samples) and Pfizer and UHK, Nat Genet 2014 (100 samples) but, of course, is smaller than that of TCGA, PanCancer Atlas (440 samples). Anyway, these numbers are not to be deemed sufficient to draw final conclusions as for distribution of somatic mutations and their clinical relevance. More grounded conclusions will probably arise as results of meta-analyses in future. We cannot guarantee unbiased enrollment of patients in our single center, but we provide an overview of clinical characteristics in table 2.

Following your advice, we compared our results with other GC databases in respect to mutual exclusivity of mutations in different genes. Indeed, we see an overlap for KMT2D and KMT2C in OncoSG, Pfizer and UHK, TCGA PanCancer Stomach Adenocarcinoma (STAD), but results of statistical analysis with correction for group numbers suggest that they pass the threshold to be considered mutual exclusive. For the analysis, we retained only the sequenced samples with mutation data (without Copy-Number Alterations) in all three datasets. For groupwise mutual exclusive test, p-values were as follows: 0.088 for Onco SG, 0.01 for TCGA STAD, and 0.5 for Pfizer study. Using the wFisher p-value combination method with sample size for each experiment, we obtained the p-value of the mutual exclusive test under the nominal significance level of 0.05. 

Another interesting observation is that when you consider missense mutations only, mutations in KMT2D and KMT2C are almost mutually exclusive in these three datasets, as they are in our study.

And the third consideration relates to the sample size under study. We have paid attention to the studies of bigger sample size, though of another cancer localization, namely, Breast Cancer METABRIC, Nature 2012 & Nat Commun 2016 (2509 samples) and Breast Cancer MSK, Cancer Cell 2018 (1918 samples), and in these datasets we witness obvious mutual exclusivity of somatic mutations in KMT2D, KMT2C and ARID1A. Although this may be a cancer type specific observation, we altogether cannot rule out sample size effect and/or peculiarities of mutation detection/interpretation in different studies.

The above considerations are now reflected in the Discussion section of the manuscript.

  1. In the materials and methods section, the authors should provide the summary of sequencing and key-parameters in mutation calling, including sequencing depth and coverage in each sample, minimum depth and VAF cutoff for mutation sites, etc.

NGS was performed with 1000x targeted sequencing depth; Torrent Variant Caller cut-off for variant allele frequency was set at 0.1, and minimum read depth of the variant allele was set at 5. These are now in the Materials and Methods section.

Designed coverage of the panel is now presented in Supplementary Tables S3 and S4.

  1. The authors claimed the mutations they identified are somatic mutations, but they did not describe whether they used normal control to distinguish somatic mutations or just based on VAF. In addition, the variant allele frequency of all mutations is essential information, must be provided.

To distinguish somatic vs germline mutations, we performed Sanger sequencing of DNA samples extracted from archived non-malignant gastric mucosa of the same patients. We now state this in “Materials and Methods”.

The variant allele frequency, total read depth, reference and variant allele read depths, etc. for each mutation are now presented in a Supplementary Table S1.

  1. In table 2, some of the numbers could not add up to the total number. For example, in section Signet ring cells, Mut+: 13+32=45, but the total number should be 49. And for total cases column, 42+89=131, not 135. The authors should check all the numbers to make sure all the results are correct. For section T/N/M, the authors should describe what are the T/N/M.

Sorry for this mistake. We have missed the last line in the table, indicating the numbers of samples with “unknown” status of signet ring cells. It is now at its place.

As for T/N/M, we have added the link to the ESMO Clinical Practice Guidelines for diagnosis, treatment and follow-up for gastric cancer to “Materials and Methods” section.

  1. Figure 3 and line 216-220. the group with mutations in the DNA methylation/demethylation genes only included 8 patients. The results from this data set is not helpful because of small patient group. Moreover, base on figure 3, all 8 patients were dead at the same time point? It doesn't make sense.

Indeed, this picture is a consequence of a small number of patients with mutations in the DNA methylation/demethylation genes, which further reduces in clinical groups. For example, figure 3(b) is based on a group of 59 patients with diffuse GC subtype, of which only one had a somatic mutation in a DNA methylation/demethylation gene, and this patient passed away within one month. As such information is not helpful, we have deleted all associations with mutations in the DNA methylation/demethylation genes from the article. In the Results we now state that the group with mutations in the DNA methylation/demethylation genes included only 8 patients and was too small to perform statistical analysis.

Minor Issues:

  1. In line 73-75, and in Figure 1, gene names should use the italic font.

Fixed.

  1. In Figure 1, the gene names can be marked with different colors to help distinguish the gene groups described in text.

Done.

  1. The authors used a designed panel that cover 25 genes, does this panel reach 100% coverage? Since the sequencing panel is an key material in this manuscript, the authors should provide the information of this panel in the supplementary.

The panel reaches 98.09% coverage by design. Note that this applies to exons and 25 bp flanking intron sequences. The information of the panel is now in Supplementary Tables S3 and S4.

  1. In table 1, column MAF: The numbers should be edited using scientific notation instead of using a lot of zeros. And there are two mutations MAF were marked as 0.000, is this a mistake or do the authors have a specific purpose? Because these two mutations doesn't have data in gnomAD.

We have edited the numbers in table 1 using scientific notation. In the original table, the presentation of numbers corresponded to that in gnomAD. Mutations with MAF marked as 0.000 were annotated in gnomAD with MAF 0.000 at the time of manuscript preparation, which surprised us as well. Now that they are absent from the database, we have indicated their MAF as “-“ in table 1.

  1. In the manuscript, the authors made several mistakes in the float number of the p-value (use "," as "."). For example, in line 211, 217, 218, 226, etc.

We have fixed this throughout the text.

Round 2

Reviewer 2 Report

In the current version of this manuscript, the author answered and fixed most of the issues brought up in the first version. I think it can be published in this journal. But before the final acceptance, there are two minor points I would like to ask the authors to address.

  1. In paragraph 344-362, the authors have done a thoroughly comparative study with other GC mutation databases. The authors should use a heatmap to illustrate these outside database mutation data of KMT2D and KMT2C. This should be a simple work but significantly improve this part.
  2. For the sequencing depth and coverage information, the authors provided the designed sequencing depth and panel design theoretical coverage. However, the depth and coverage of real data in each sample is very important quality control information. Please provide a supplemental table with mapped data depth and coverage for each sample.

Last, the pdf file generated by the MDPI system seems to retain the original text and revised text, but did not show the deletion line like the way Word software does. Please double-check the sentences and format around the revised part, since I can't help to check with this pdf file.

Author Response

Dear Reviewer,

Thank you for your advice.

We have addressed your two points as follows:

  1. In paragraph 344-362, the authors have done a thoroughly comparative study with other GC mutation databases. The authors should use a heatmap to illustrate these outside database mutation data of KMT2D and KMT2C. This should be a simple work but significantly improve this part.

We have added Figure 6 illustrating analysis of mutual exclusivity of KMT2D and KMT2C mutations on the data presented in this article and in other gastric cancer mutation databases.

  1. For the sequencing depth and coverage information, the authors provided the designed sequencing depth and panel design theoretical coverage. However, the depth and coverage of real data in each sample is very important quality control information. Please provide a supplemental table with mapped data depth and coverage for each sample.

We have added Supplementary Table S5 to present this information.